# *Stenamoeba dejonckheerei* sp. nov., a Free-Living Amoeba Isolated from a Thermal Spring

**DOI:** 10.3390/pathogens9070586

**Published:** 2020-07-17

**Authors:** Manuel Alejandro Borquez-Román, Luis Fernando Lares-Jiménez, Libia Zulema Rodriguez-Anaya, Jose Reyes Gonzalez-Galaviz, Paul A. Fuerst, José Cuauhtémoc Ibarra-Gámez, Ramón Casillas-Hernández, Fernando Lares-Villa

**Affiliations:** 1Programa de Doctorado en Ciencias en Biotecnología, Instituto Tecnológico de Sonora, Ciudad Obregón, Sonora 85000, Mexico; manuel-177@hotmail.com; 2Departamento de Ciencias Agronómicas y Veterinarias, Instituto Tecnológico de Sonora, Ciudad Obregón, Sonora 85000, Mexico; luis.lares@itson.edu.mx (L.F.L.-J.); jose.ibarra@itson.edu.mx (J.C.I.-G.); rcasillas@itson.edu.mx (R.C.-H.); 3CONACYT—Instituto Tecnológico de Sonora, Ciudad Obregón, Sonora 85000, Mexico; libia.rodriguez@conacyt.mx (L.Z.R.-A.); jose.gonzalez@itson.edu.mx (J.R.G.-G.); 4Department of Evolution, Ecology and Organismal Biology, The Ohio State University, Columbus, OH 43210, USA; fuerst.1@osu.edu

**Keywords:** amoebae, free-living amoebae, thermophilic, *Platyamoeba*, *Naegleria*, *Acanthamoeba*, *Balamuthia*, *Stenamoeba*, distribution, environment

## Abstract

Two amoeboid organisms were obtained from water samples taken from a thermal spring, "Agua Caliente", in Northwestern Mexico. The isolates were obtained when samples were cultivated at 37 °C on non-nutrient agar coated with *Escherichia coli.* The initial identification of the isolates was performed morphologically using light microscopy. The samples were found to have trophozoite morphology consistent with members of the genus *Stenamoeba*, a genus derived in 2007 from within the abolished polyphyletic genus *Platyamoeba*. Further analysis was performed by sequencing PCR products obtained using universal eukaryotic primers for the small subunit ribosomal ribonucleic acid (SSU rRNA) gene. Sequencing primers were designed to allow the comparison of the 18S rRNA gene sequences of the new isolates with previous sequences reported for *Stenamoeba.* Phylogenetic relationships among sequences from *Stenamoeba* were determined using Maximum Likelihood analysis. The results showed the two "Agua Caliente" sequences to be closely related, while clearly separating them from those of other *Stenamoeba* taxa. The degrees of sequence differentiation from other taxa were considered sufficient to allow us to propose that the Mexican isolates represent a new species.

## 1. Introduction

Studies were conducted in February 2017 to identify the occurrence of potentially pathogenic free-living amoebae within a natural hot spring—two isolates, in particular, received our attention. These isolates were morphologically identified as potential members of the genus *Stenamoeba* [1,2]. Further analysis, detailed here, has revealed that these isolates represent a new member of *Stenamoeba*, genetically differentiated from any previously reported form. The genus *Stenamoeba* was originally defined in 2007 [3], where it was shown that the taxa previously denoted *Platyamoeba stenopodia*, Page 1969, was not correctly placed within the Vanellida, but was a member of the Thecamoebida [3,4]. The new genus till then considered monospecific (*Stenamoeba stenopodia*), soon expanded with the identification of a number of new species [3,5,6,7]. To date, six species of the genus *Stenamoeba* have been described; *S. stenopodia*, *S. amazonica*, *S. limacina*, *S. berchidia*, *S. sardiniensis*, and *S. polymorpha*. *S. stenopodia*, the former *Platyamoeba stenopodia*, was isolated from a lake in Alabama, USA [3,8], *S. limacina* and *S. amazonica* were isolated from kidney tissue and gill tissue from fishes of the Czech Republic and Peru, respectively [5], *Stenamoeba sardiniensis* and *S. berchidia* were described from a single soil sample in Sardinia, Italy [6], and *S. polymorpha*, was isolated from the diarrheic stool of a domesticated horse in Great Falls Virginia, USA [7]. Here, we show that the two amoebae isolates from the “Agua Caliente” hot springs are closely related, and that they represent a novel, previously undescribed species within *Stenamoeba*, well-differentiated genetically from previously defined taxa, and which we propose to represent as *Stenamoeba dejonckheerei* sp. nov.

## 2. Results

### 2.1. Growth Characteristics

All attempts to axenize the cultures of M32 and M33, using different procedures and culture media were unsuccessful. Differences in the growth rate and appearance when strains M32 and M33 were grown in non-nutritive agar plates added with *Escherichia coli* (NNE), at 37 °C were observed. Strain 33 grew slower than strain 32 when incubated at 37 °C, and also strain 33 tended to grow buried in the agar. Both strains did not grow at 45 °C.

### 2.2. Measurements and Morphology by Light Microscopy

The motile trophozoites have an elongated form shape, with a round and frontal hyaline area (Figure 1I–K), and occasionally exhibit a nearly fan-shaped appearance on agar plates (Figure 1G,H). The hyaloplasm is very pronounced, covering up to 2/3 of the cell when it is in fan-shaped appearance and only 1/3 when it has an elongated shape. The granular part of the cell is narrower than the hyaline part, often pointed at the posterior end. An uroidal structure is observed and a single contractile vacuole that is also located in the posterior part of the cell. Locomotion could be monotactic or polytactic, with flowing hyaloplasm leading the direction of movement. Amoebae in movement ranged in length from 15 to 24 µm (mean 20.3 ± 2.7 µm). For the same amoebae, measurements of the width ranged from 4 to 11 µm (mean 6.04 ± 2.03 µm). The length to width ratio (L/W) of these cells ranged from 1.4 to 5.3 (mean 3.7). Non-directionally moving cells adopted variable shapes, often forming dactylopodia-like projections (Figure 1D,E). A single vesicular nucleus was always located close to the border between the hyaloplasm and the granuloplasm, and the diameter of the nucleus ranged from 1.4–5.29 µm (mean 2.57 ± 0.35 µm). Floating amoeba produced forms with emerging bifurcating long, slender, mostly hyaline, and not pointed pseudopods (Figure 1A–C). Amoebae formed smooth, spherical, double-walled cysts that ranged in diameter from 6.5 to 10 µm (mean 7.84 ± 1.22 µm) (Figure 1F).

### 2.3. Molecular Characterization of the M32 and M33 Strains

PCR products were obtained by using 18S universal PCR primers (as described in the Materials and Methods section, below). Both isolates yielded similar sized products, estimated to be approximately 2100 bp in size (Figure 2). This product was used for subsequent DNA sequencing.

When the PCR products were sequenced, only sequences of 1700 bp were obtained, due to the lack of sequencer capacity. These sequences were subsequently compared with 18S rRNA sequences in the GenBank international database [9]. When this comparison was performed, the closest similarity was found to be with *Stenamoeba berchidia*, but with substantial divergence, yielding sequence identity of only 91%. Thus, strains M32 and M33 were initially considered to represent an unidentified taxon, i.e., *Stenamoeba* sp.

For further analysis, the sequences from M32 and M33 were compared with several 18S rRNA sequences from various *Stenamoeba* taxa that were present in GenBank. This allowed a search for additional PCR primers that would likely be able to amplify smaller segments from any of the isolates that are in the database. *Stenamoeba* internal primers were designed from the FASTA alignment file of eight 18S rRNA gene sequences attributed to *Stenamoeba* that exceeded 1800 bases in length, and included a sequence called "primer amoeba", which was a rough consensus sequence of the conserved parts of the other smaller sequences. Seven pairs of primers (St1–St7) with short sequences were identified that would produce products that span the length of the sequences of all the *Stenamoeba* species. A gradient PCR was used to find the optimal alignment temperature for each primer pair.

Some regions turned out to be more difficult to amplify. Consequently, two additional primers (StM1 and StM2) were developed from the *Stenamoeba berchidia* 18S rRNA sequence (KF547922), because this sequence showed greater similarity with the M32 and M33 isolates. These primers, designed with the help of the Primer3plus software (http://www.bioinformatics.nl/primer3plus) [10], were developed to replace previous primers, which produced amplification problems. With the design of primers StM1 and StM2, it was possible to obtain an almost complete sequence of the samples from M32 and M33, with the exception of a missing area of ~170 bp. We assumed that these primers were successful because they were designed from the sequence of a single species. Therefore, an added primer (StA1) was designed with the help of the Qiagen CLC Genomics Workbench v.20.0.3 software (Qiagen, Redwood City, CA, USA), from the conserved region that is present in the 18S rRNA gene sequences of all the species of the genus *Stenamoeba* in GenBank. This would allow us to cover the missing region, hoping to obtain the continuous complete sequence for two samples. The set of primers used to genetically characterize the M32, and M33 strains are shown in Table 1. Sequences for the two isolates have been deposited in GenBank. 

The sequence obtained for M32 was 2029 bases in length (GenBank accession #MT386405), while the sequence for M33 does not include the first 53 bases, due to sequencing problems, yielding a 1976 base long sequence (acc #MT386374).

### 2.4. Alignments and rRNA Secondary Structure

When aligning the assembled M32 and M33 samples against other *Stenamoeba* species strains deposited in GenBank, a number of potentially hypervariable regions associated with insertion/deletion (in/del) events could be observed. These regions are associated with various stem/loop regions in the 18S rRNA secondary structure. A general description of the helix structure and location of more variable positions within eukaryotic small subunit rRNAs was described by Neefs et al. [12]. The regions of in/del within *Stenamoeba* 18S rRNA sequences make accurate alignment along the entire gene problematic, primarily because of the uncertainty of the homology of different nucleotide sites between different taxa. The in/del regions of *Stenamoeba* are listed in Table 2, located by reference to the nucleotide position in the 18S rRNA sequence of M32 (MT386405), and the helix and eukaryotic variable regions in which they are located.

Two examples illustrating aspects of the hypervariable regions could be given. In the first, an example of the secondary structure that is involved in these regions is provided in Figure 3.

The secondary structure from *S. polymorpha*, analogous to that shown in Figure 3, was compared with other *Stenamoeba* taxa as part of the argument for new species designation for *S. polymorpha*. The structure in Figure 3 can be compared with those in Figure 4 of Peglar et al. [7], which provides the equivalent structures from eight sequences from *Stenamoeba* taxa. The structure in Figure 3, should, and does, most closely match that of *S. berchidia*. Four single base substitutions, and a single in/del have occurred that differentiate the *S. berchidia* sequence from that in M32 and M33, but the secondary structure remains intact, with the base substitutions occurring in compensating manners within the stems, and the in/del occurring in an unpaired portion of the structure. Unfortunately, in the paper of Peglar et al. [7], the *S. berchidia* structure is mislabeled as *S. sardiniensis,* a mistake that could lead to confusion.

A second example of the type of in/del variation seen when considering the position of the M32/M33 form with other members of *Stenamoeba* is shown in Figure 4. The M32 and M33 samples are compared against the *Stenamoeba* species strains deposited in GenBank. The hypervariable region shown in Figure 4 is region 4 listed in Table 2. This is a hypervariable region of ~180. Within this region, a ~100 bp deletion occurs that is unique to our samples (Figure 3).

The variable regions that are dominated by in/del sites contribute greatly to the understanding of the divergence of the taxa of *Stenamoeba* for which information is available. We have examined the sizes of these variable regions in each of the sequences that we have studied. The results are shown in Table 3. As mentioned above, it is clear that for the new sequences of M32 and M33, region 4 showed the greatest contrast with all the other species of *Stenamoeba*. For other regions, M32 and M33 were similar, although not always identical in size to the regions in *S. berchidia*.

### 2.5. Phylogenetic Analysis of Sequences of Stenamoeba

The aligned sequences from nine other taxa attributed to *Stenamoeba* were compared to the sequences from M32 and M33, using the Maximum Likelihood (ML) method, as implemented in MEGAX [13]. The importance of the hypervariable regions discussed above requires careful analysis of phylogenetic relationships. ML trees were constructed using two datasets. One set included all sites, considering a deletion as a separate pseudo-nucleotide type for the analysis. The second set included a censored collection of sites excluding all in/del sites for which any single sequence was represented as a deletion. Differences between the trees will be discussed. Both alignments begin at position 54 of the M32 sequence.

The results of the ML analysis for all the sites are presented in Figure 5. The tree has been rooted using the sequence of *Paradermamoeba levis* (JN247435); *Stenamoeba* is classified within the Thecamoebida [3,4], but shows substantial differentiation from other thecamoebid taxa. As expected, Figure 5 shows that the sequences of isolates M32 and M33 are found to be closest to the sequence of *S. berchidia*.

Figure 5 indicates that *Stenamoeba* is separated into two groups, one containing the sequences of M32 and M33, together with three species, *S. berchidia*, *S. limacina*, and *S. polymorpha*. The second group contains the type of species for the genus *S. stenopodia* together with *S. amazonica*, and several unidentified *Stenamoeba* sp. isolates. The bootstrap values from the tree suggest that the differentiation between most of the sequences in the analysis is likely to be significant.

The ML tree obtained from the censored dataset has the same topology. The only differences involved slight lowering of the bootstrap support for various groupings, an indication of the importance of the variable segments in determining the relationship among taxa.

One of the named species of *Stenamoeba, S. sardiniensis*, is not shown on the tree. The 18S rRNA gene sequence of *S. sardiniensis,* reported in 2014, is a partial sequence, spanning only 772 bases. The species description was not based only on the molecular data, which suggests that *S. sardiniensis* is close to *S. stenopodia*.

Although the ML trees show significant differentiation between sequences, is this sufficient to assume that the sequences of the M32 and M33 isolates might represent a new, previously unrecognized species within *Stenamoeba*? To examine this further, we analyzed the degree of sequence similarity, or its inverse, sequence differentiation. The results of our studies are given in Table 4. As with the estimation of the phylogenetic tree, estimates of sequence differentiation depended on whether the in/del sites were included in the calculation. Excluding the in/del sites, the average evolutionary divergence between the five named species in the analysis averaged 0.093 (range 0.072–0.111). The M32 and M33 isolates average of 0.042 censored divergence from *S. berchidia*. When the in/del sites were included, the evolutionary divergence between the named species averaged 0.155 (range 0.104–0.200). The M32 and M33 isolates averaged 0.114 total divergence from *S. berchidia*. Given the unique sequence characteristics of the V4 region of the 18s rRNA sequences of M32 and M33, and the degree of total sequence divergence, the two isolates from Sonora appeared to represent a taxon of *Stenamoeba* worthy of species-level designation.

Since the recognition that the single-species *Stenamoeba* clade represented a distinct genus of amoebae [4], the number of well-defined taxa with species names has increased substantially. From the description of the original single species, *S. stenopodia*, which justified the new genus, *Stenamoeba* has now expanded to include five additional named species. There also exist a number of isolates that show sequence differentiation for the 18S rRNA gene at levels that are equivalent to those seen between the named species, as shown in Table 4.

In addition, a set of environmental uncultured eukaryotic clones were identified in studies of biofilms [14]. This set included more than 800 protist clones, most of which appeared to represent samples of *Vermamoeba vermiformis*. However, in comparison with other protist eukaryotic sequences, 47 of the clones showed the greatest similarity to *Stenamoeba* sequences, and specifically with the M32 and M33 sequence isolates reported here, followed by the sequence of *S. berchidia*. The sequence comparisons with these uncultured clones occurred over ~650 bases in the 5′ end of the sequence. Despite showing over 98.5% sequence similarity to M32 and M33, the clones contained distinct differences in the *Stenamoeba* variable regions in the V2 portion of the eukaryotic 18S rRNA, which occurred between base 267 and base 328 of the M32 sequence. These clones appear to represent at least one (or more) taxon, closely related to but not identical to the sequences described here.

## 3. Discussion

Since their isolation at 37 °C, the M32 and M33 strains, subsequently identified as members of the genus *Stenamoeba* by Lares-Jiménez et al. [2], were kept at room temperature in a monoxenic culture (NNE). When attempts were made to grow the cultures at 37 °C and 45 °C, growth occurred at the lower temperature, but the isolates did not grow at 45 °C, although the water temperature at the time of sampling was 45.5 °C. One possible explanation is that these organisms were thermotolerant, but non-thermophilic, so they did not grow at 45 °C.

Other isolates within the genus *Stenamoeba* have been observed to have various relationships with water temperature. *S. stenopodia* was isolated at −17 °C [15]. A strain of *Stenamoeba* sp. isolated from a thermal bath in Swiss spas grew at 18, 37, and 42 °C [16]. The *Stenamoeba* sp. isolate PC:10/15 P137C_St (accession #MT109100) was sampled from an Italian geothermal hot spring. It could be deduced that amoebae of the genus *Stenamoeba*, in addition to being thermotolerant, are also eurythermal organisms.

To carry out pathogenicity tests, several attempts were made to obtain axenic cultures with common media used for FLA. Cerva’s medium, which is excellent for the growth of *Naegleria* spp., and *Acanthamoeba* spp., and the BM-ITSON and BM-3 media in which *B. mandrillaris* grows very well [17,18,19]. The results of all attempts were unsuccessful.

The morphological characteristics found in the studied strain share some characteristics with the previously described species of the genus *Stenamoeba* [5,6,7,15]. If we compare the measurements of the trophozoite of the M32 strain with the six *Stenamoeba* species described in the literature, we find that it is among the longest, only surpassed in size by *S. polymorpha* [7]. As for the width, it is very similar to *S. limacina*, with a measurement of almost twice that of *S. sardiniensis*, which is the narrowest species [5,6]. As for the L/B ratio, the M32 strain also resembles *S. limacina*, although the highest ratio is presented by *S. polymorpha* [5,6,7]. An interesting feature of the M32 strain is that it has an uroid-like structure in the anterior region of the elongated and fan shapes, which is not reported in the previously described species and a finer study is necessary to determine the type of uroid according to Page’s classification [20]. Although single layer cysts have been reported for *S. amazonica* and *S limacine* [5,6], for *S. stenopodia*, *S. sardiniensis*, *S. Berchidia*, and *S. polymorpha*, the presence of two layers in the cyst appears to be found when the cyst it is in a mature state [7]. However, since in the last four species mentioned, as well as in the morphology of the cyst for the M32 strain, the description was made by light microscopy, it is necessary to perform electron microscopy to confirm or rule out the presence of a second layer. The same premise above applies to “ostiole”, which although the structure is not mentioned in *Stenamoeba*’s species descriptions, Page writes for *Platyamoeba stenopodia*’s description, "Excystment by bursting an irregular hole in the cyst wall" [8]. The presence of empty cysts in the culture medium is an indication of the type of amoeba excystment process.

Seven primer sets were designed and tested for the first time (Table 1), but only primer sets St1, St6, and St7 successfully produced amplicons, with sizes 420, 316, and 846 bp, respectively. Primer sets St2, St3, St4, and St5 did not amplify any regions of M32 or M33. Subsequently, primer sets StM1 and StM2 (Table 1) were designed with degenerate nucleotides in regions prone to single nucleotide polymorphisms (SNPs). Once all regions of both M32 and M33 samples were successfully amplified, the fragments were purified and sequenced in both directions. By aligning the M32 and M33 assemblies against other *Stenamoeba* species, variable regions characterizing this genus were observed, and a 98 bp deletion exclusive to our strains was identified (Figure 4). This necessitated a final set of oligonucleotide primers, StA1 (Table 1), which amplified segments within the regions corresponding to the StM1 and StM2 primer sets.

DNA analysis of the two isolates produced nucleotide sequences of 2029 and 1976 in length, which differed by a single nucleotide in the overlap of 1976 bases. The sequences were confirmed to be more closely related to the sequences attributed to the members of *Stenamoeba* (Figure 5). Phylogenetic relationships between *Stenamoeba* sequences were determined by maximum likelihood analysis and showed that the two "Agua Caliente" sequences were closely related to each other while being clearly separated from other taxa of *Stenamoeba*. Phylogenetic analysis indicates that *Stenamoeba berchidia* is the closest known relative of the isolates (Figure 5 and Table 4). The degrees of sequence differentiation from other taxa were considered sufficient to allow us to propose that the Mexican isolates represented a new species.

Given our findings, it was decided to propose a new species, *Stenamoeba dejonckheerei,* named in honor of Dr. Johan F. De Jonckheere, for his great contribution to the research of free-living amoebae over the years.

## 4. Taxonomic Conclusions

Based on the combination of morphological and molecular data, a new species of the genus *Stenamoeba* Smirnov, Nassonova, Chao et Cavalier-Smith [3], is described.

### Stenamoeba dejonckheerei sp. n.

With *Stenamoeba* characteristics.

Trophozoite: The motile trophozoites have an elongated form shape, with a round and hyaline anterior part and occasionally exhibiting a nearly fan-shaped appearance on agar plates. The hyaloplasm is very pronounced, covering up to 2/3 of the cell when it is in a fan-shaped appearance and only 1/3 when it has an elongated shape. The granular part of the cell is narrower than the hyaline part, often pointed at the posterior end. An uroidal structure is observed and a single contractile vacuole that is also located in the posterior part of the cell. Locomotion could be monotactic or polytactic, with flowing hyaloplasm leading the direction of movement. Amoebae in movement ranged in length from 15 to 24 µm (mean 20.3 ± 2.7 µm), and of width ranged from 4 to 11 µm (mean 6.04 ± 2.03 µm). The length to width ratio (L/W) ranged from 1.4 to 5.3 (mean 3.7). A single, vesicular nucleus was always located close to the border between the hyaloplasm and the granuloplasm, and the diameter of the nucleus ranged from 1.4–5.29 µm (mean 2.57 ± 0.35 µm). Floating amoeba produced several small pseudopods-like fingers around the cell. Amoebae formed smooth, spherical, double-walled cysts that ranged in diameter from 6.5 to 10 µm (mean 7.84 ± 1.22 µm).

Food: bacterivorous.

Observed habitat: fresh water; thermotolerant to at least 45.5 °C.

Growth in vitro. Maintained in Petri dishes with non-nutrient agar supplemented with a gram-negative bacterium such as live *Escherichia coli* (NNE), at room temperature. Grows well at 37 °C on an agar plate.

A distinct short nucleotide pattern in region V4 differentiates this species from any other known *Stenamoeba*. Unique SSU-rRNA secondary structure predicted for V8 helices 45 and 46.

Gene sequence data: GenBank Accession ID: MT386405.

Type locality: “Agua Caliente”, a natural thermal water spring, in Sonora State, Mexico, with geographic coordinates (27.732079°–109.836395°).

Etymology: the species name refers to Dr. Johan F. De Jonckheere.

Type material: Type culture, *Stenamoeba dejonckheerei* sp. n. strain M32 is maintained at Laboratorio de Biología Molecular del Instituto Tecnológico de Sonora; there is currently no strain of available at either ATCC or CCAP.

## 5. Materials and Methods

### 5.1. Growth Characteristics

The two isolates identified as *Stenamoeba* were designated strains M32 and M33 and were maintained in Petri dishes with non-nutrient agar coated with live *Escherichia coli* (NNE), at room temperature, until their preparation for DNA extraction and molecular identification. At that time, when increased growth was required for DNA extraction, the temperature of cultures was raised and maintained at 37 °C. Various attempts were made to axenize the cultures using different procedures and culture media (for the purpose of carrying out pathogenicity tests), including Cerva’s medium, BMI, and BM-3 media, and procedures such as using liquid media with dead *E. coli* [17,18,19].

### 5.2. Light Microscopic Characterization

Observations and measurements of live cultures of strain M32 in Petri dishes with NNE, were made using an inverted microscope Axiovert 135 ZEISS (Carl Zeiss, Gottingen, Germany). Observations and photographs of amoebae moving across the glass surfaces were done using a microscope Axiolab ZEISS (Carl Zeiss, Gottingen, Germany). Digital photomicrographs of cysts, locomotive, and amoebae stationary were obtained using a SONY camera adapted to the microscopes and software for video capture ZOGIS Real Angel 220 PCI. Fifty measurements of cysts and trophozoites were made using stage and eyepiece micrometers. Several different slide preparations were used, including wet mounts and semi-permanent preparations, these last fixing the coverslip with nail polish.

### 5.3. Molecular Characterization of the M32 and M33 Strains

To better determine the identity of the two isolates, molecular analysis was performed. The DNA was extracted from amoebae growing on monoxenic cultures, using the DNeasy extraction kit (Qiagen, Inc., Valencia, CA, USA), followed by measurement of the DNA concentration with a NanoDrop 2000c spectrophotometer (Thermo Scientific, Wilmington, DE, USA). PCR was initially performed with the GoTaq Flexi DNA Polymerase kit (Promega, Madison, WI, USA) using universal 18S rRNA primers that have been used to successfully identify other amoebic genera. The universal primers used for eukaryotes were ERIB1 and ERIB10 [11] (Table 1).

PCR products were visualized on 1% agarose gel stained with ethidium bromide on a UV transilluminator (Figure 2). The PCR products were purified and sequenced by the Sanger method. The identification of the sequences was performed by a BLASTn search of the GenBank database. This yielded best matches to deposited sequences assigned to taxa of *Stenamoeba*. The primers were subsequently designed using the Primer3Plus software (http://www.bioinformatics.nl/primer3plus) [10]. Six sets of primers were ultimately utilized to obtain the full-length 18S rRNA gene sequence from the two isolates (Table 1). Primers were designed to produce overlapping regions of amplification to allow the assembly of the 18S rDNA sequence. Assembly was facilitated using the CAP3 software (http://doua.prabi.fr/software/cap3) [21].

All the tested PCR protocols were performed with 35 cycles of amplification. The new primers underwent a standardization process, and once the PCR products were obtained, they were purified and sequenced in triplicate by the Sanger method, sending duplicate samples to Laboratory of Integral Diagnosis of Animal Pathology of the Technological Institute of Sonora and to the Synthesis and DNA Sequencing of the Institute of Biotechnology, National Autonomous University of Mexico.

### 5.4. Alignments and Phylogenetic Analysis

The nucleotide sequences obtained for each primer by triplicate were assembled using the CAP3 software. Alignments were made with the CLC Genomics Workbench v.20.0.3 platform (https://digitalinsights.qiagen.com) by adding the sequences deposited in the GenBank and classified as complete or almost complete of various *Stenamoeba* species and strains (KF547922, EU377587, GU810183, KU955320, JQ271721, GU810184, AY294144, JX312796, KF547924, KF547921).

A phylogenetic tree (Figure 5) was created with the MEGAX software developed by The Pennsylvania State University (https://www.megasoftware.net/) [12]. Maximum Likelihood analysis was performed. The parameters for the ML analysis, identified using the Best Model option, were the T92+G (Tamura three-parameter model using a discrete Gamma distribution. A heuristic search approach was obtained automatically by applying Neighbor-Join and BioNJ algorithms to an estimated pairwise distance matrix using the Maximum Composite Likelihood (MCL) approach. Bootstrapping with 500 repetitions was used to determine the confidence of clades.

Evolutionary divergence was calculated for a censored dataset, with the in/del sites removed, by using the Compute Pairwise Distance application within MEGAX. Divergence incorporating the in/del sites was calculated by a pairwise comparison of conserved and variable sites in a pairwise alignment of sequences in MEGAX.

## Figures and Tables

**Figure 1 pathogens-09-00586-f001:**
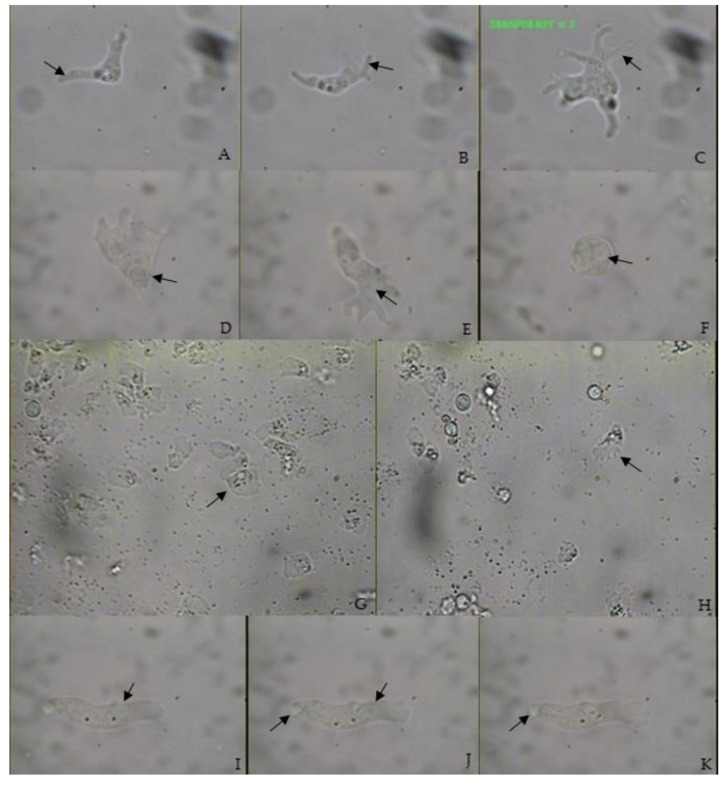
(**A**–**C**) Floating forms with emerging bifurcating long, slender, mostly hyaline, and not pointed pseudopods (1000×). (**D**,**E**) Amoebae moving across the glass surfaces showing a contractile vacuole, in the posterior region and nucleus, near the border between the hyaloplasm and the granuloplasm (1000×). (**F**) A Cyst where the nucleus can be distinguished (1000×). (**G**,**H**) Trophozoites on Petri dishes with non-nutritive agar plates added with *Escherichia coli* (NNE), showing a fan-shaped appearance (400×). (**I**–**K**) Trophozoites with meandering motion like “limax” (that form a single broad anterior pseudopodium and flow forward sluggishly), showing an elongated form shape with an uroidal structure, and nucleus near the border between the hyaloplasm and the granuloplasm (1000×).

**Figure 2 pathogens-09-00586-f002:**
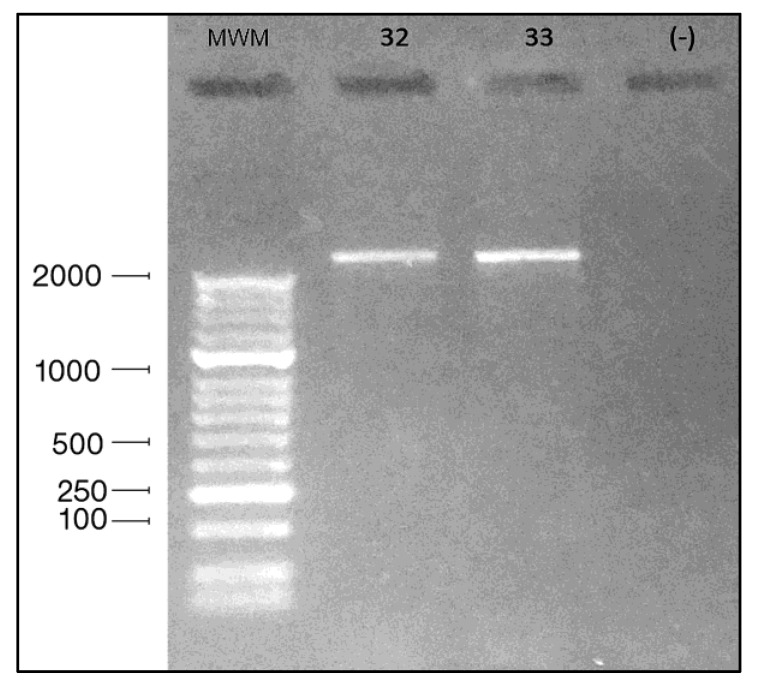
The initial PCR product for the 18S rRNA gene from isolates M32 and M33. Gel is 1% agarose, stained with ethidium bromide. Molecular weight determined with HyperLadder50bp™ (Bioline Meridian Biosciences, Memphis, TN, USA).

**Figure 3 pathogens-09-00586-f003:**
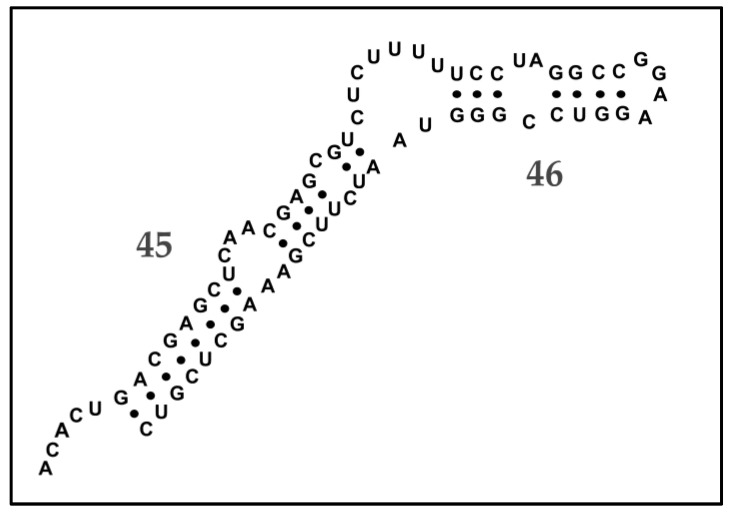
Predicted SSU-rRNA secondary structure for helices 45–46 contains motifs that discriminate known members of *Stenamoeba*. Adapted from Peglar et al. [7].

**Figure 4 pathogens-09-00586-f004:**
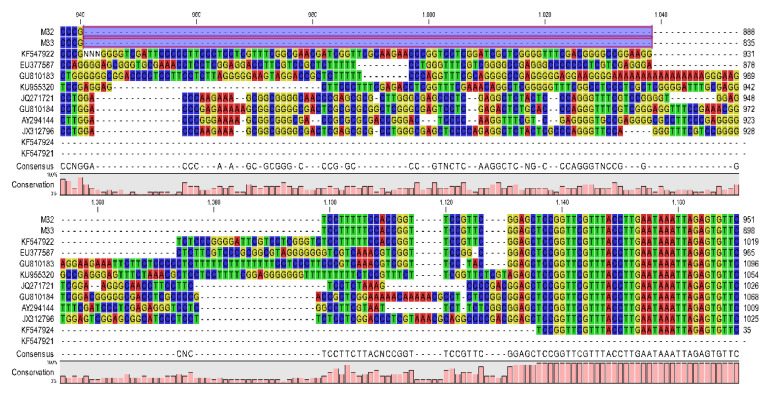
Alignment of a characteristic hypervariable region occurring in all representatives of *Stenamoeba*. The blue box marks the variable region corresponding to samples M32 and M33.

**Figure 5 pathogens-09-00586-f005:**
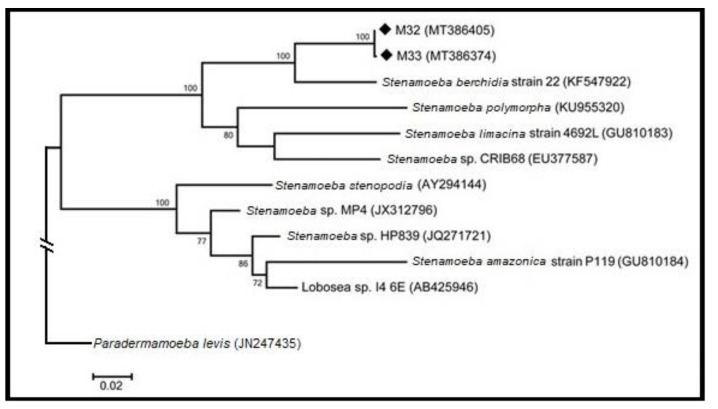
Maximum likelihood tree for all sites Phylogenetic tree of the genus *Stenamoeba*.

**Table 1 pathogens-09-00586-t001:** Set of primers used to genetically characterize the M32 and M33 strains.

Primer	5′–3′	Protocol	Amplicon
ERIB1- ERIB10	F- ACCTGGTTGATCCTGCCAG R- CTTCCGCTGGTTCACCTACGG	[11]	2100 bp
St1	F- CGGTGAAACTGCGAATGGCTC R- GGAGCCTGAGAAACGGCTAC	95 °C (5 min), 95 °C (1 min), 64 °C (30 s), 72 °C (30 s), 72 °C (5 min)	420 bp
StM1	F- CGGGTGACGGAGAATTAGGG R- CGGTCCTAGAAACCAACGAA	95 °C (5 min), 95 °C (1 min), 62 °C (1 min), 72 °C (1 min), 72 °C (5 min)	715 bp
StA1	F- ACRATTGGAGGGCAAGTCTG R- GCAAATGCTTTCGCTGAAGT	95 °C (5 min), 95 °C (1 min), 60 °C (45 s), 72 °C (45 s), 72 °C (5 min)	700 bp
StM2	F- CACACGCCYRGATACTTTAG R- GACTATGCAATCCCAGTGCA	95 °C (5 min), 95 °C (1 min), 62 °C (1 min), 72 °C (1 min), 72 °C (5 min)	733 bp
St6	F- CCTCTGGTGRRGTTCGTAGC R- TCGCTCCTACCGATTGAACG	95 °C (5 min), 95 °C (1 min), 63 °C (30 s), 72 °C (30 s), 72 °C (5 min).	316 bp
St7	F- GCGAAAGCATTTGCCAAGGA R- CGCTCCTACCGATTGAACGRTCCGGTGAA	95 °C (5 min), 95 °C (1 min), 55 °C (1 min), 72 °C (1 min), 72 °C (5 min).	846 bp

F—forward, R—reverse.

**Table 2 pathogens-09-00586-t002:** Regions of insertion/deletion difference within sequences of *Stenamoeba*, relative to the location within the 18S rRNA gene sequence of the M32 isolate.

Regions	Location	Helix Location *	Variable Region *
region 1	191–208	10	V2
region 2	267–328	10–11	V2
region 3	744–787	E23	V4
region 4	888–905	E23-6-E23-7	V4
region 5	1566–1625	43	V7
region 6	1769–1780	45–46	V8

* Helix and Variable region positions are assigned following the format of Neefs et al. [12].

**Table 3 pathogens-09-00586-t003:** Size of variable regions (in bases) in each species/taxa of *Stenamoeba*. Regions are those defined in Table 2.

Species/Region	1	2	3	4	5	6
M32 (MT386405)	16	61	42	16	58	10
M33 (MT386374)	16	61	42	16	58	10
*S. berchidia* str. 22 (KF547922)	16	64	38	139	69	9
*S. limacina* str. 4692L (GU810183)	13	60	42	165	77	10
*S. polymorpha* (KU955320)	15	64	61	142	68	9
*Stenamoeba* sp. CRIB68 (EU377587)	13	61	15	129	73	9
*S. amazonica* str. P119 (GU810184)	32	57	17	143	79	67
*S. stenopodia* (AY294144)	33	36	20	126	82	10
*Stenamoeba* sp. MP4 (JX312796)	33	50	20	140	79	34
*Stenamoeba* sp. HP839 (JQ271721)	29	48	18	114	79	34
*Lobosea* sp. I4 6E (AB425946)	29	46	23	151	79	34

**Table 4 pathogens-09-00586-t004:** Evolutionary divergence among the 18S rRNA gene sequences from taxa of *Stenamoeba*. The values above the diagonal are pairwise nucleotide divergence (differences/length of sequence) for all sites, including sites with in/del differences. The sequence similarity is calculated as (1—divergence). The values below the diagonal represent evolutionary divergence for the censored dataset of the nucleotide sites in which sites involving in/dels in any of the eleven taxa were removed.

Taxa	Species	1	2	3	4	5	6	7	8	9	10	11
1	M32 (MT386405)	-	0.001	0.113	0.151	0.146	0.135	0.194	0.171	0.179	0.168	0.176
2	M33 (MT386374)	0.001	-	0.114	0.152	0.147	0.136	0.196	0.173	0.180	0.169	0.178
3	*S. berchidia* str. 22 (KF547922)	0.041	0.042	-	0.130	0.128	0.110	0.179	0.153	0.156	0.154	0.164
4	*S. limacina* str. 4692L (GU810183)	0.076	0.073	0.082	-	0.126	0.112	0.200	0.177	0.176	0.170	0.173
5	*S. polymorpha* (KU955320)	0.064	0.065	0.087	0.077	-	0.114	0.186	0.170	0.168	0.170	0.169
6	*Stenamoeba* sp. CRIB68 (EU377587)	0.063	0.063	0.073	0.069	0.077	-	0.167	0.146	0.155	0.151	0.156
7	*S. amazonica* P119 (GU810184)	0.089	0.086	0.102	0.111	0.108	0.103	-	0.104	0.081	0.071	0.077
8	*S. stenopodia* (AY294144)	0.081	0.079	0.090	0.097	0.101	0.095	0.072	-	0.069	0.080	0.091
9	*Stenamoeba* sp. MP4 (JX312796)	0.079	0.078	0.093	0.102	0.097	0.098	0.063	0.052	-	0.039	0.046
10	*Stenamoeba* sp. HP839 (JQ271721)	0.079	0.076	0.097	0.098	0.097	0.093	0.051	0.050	0.031	-	0.040
11	*Lobosea* sp. I4 6E (AB425946)	0.084	0.080	0.097	0.104	0.102	0.102	0.057	0.065	0.036	0.026	-

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
