# Peer review of "Stenamoeba dejonckheerei sp. nov., a Free-Living Amoeba Isolated from a Thermal Spring"

_pathogens, 2020, doi:10.3390/pathogens9070586_

Round 1
Reviewer 1 Report
This is a well-written manuscript with very interesting findings. This article studied the isolation of Stenamoeba dejonckheerei sp. nov., a free-living amoeba isolated from a thermal spring in Northwestern Mexico. I have a few minor comments to improve the manuscript. Below are my specific comments:
- Lines 44: Please cite appropriate references if possible.
- Please discuss the Stenamoeba in the introduction section (such as habitats, and other species) to signify the importance of the current work.
- Line 55-58: Please avoid the repetition of the materials and methods.
- Line 187 and Line 211: Please provide appropriate references.
- Line 225: The opening of the line should be changed.
- Line 229-230: Please cite an appropriate source and it might be helpful to compare this characteristic with other species of Stenamoeba to justify the claim.
- Line 234: What is the source? Cite an appropriate source please.
- Line 306: It should be section 5.1
- Line 321: Please change to 5.2
- Line 333: Please provide a web link for the Primer3Plus software.
- Line 336: Same as the previous comment (provide a web link).
- Line 348: Please provide the name of the developer and a weblink for the MEGAX software
- Line 350: Please cite an appropriate source.
Author Response
I am sending the revised manuscript No. Pathogens-844572, titled Stenamoeba dejonckheerei sp. nov., a Free-Living Amoeba Isolated from a Thermal Spring, once the reviewers suggestions were attended point by point as can be read in the following lines.
To Reviewer 1
1) Lines 44: Please cite appropriate references if possible. Please discuss the Stenamoeba in the introduction section (such as habitats, and other species) to signify the importance of the current work.
Answer. The suggestion was accepted and the introduction was improved by adding information on the species, habitats and sources of isolation.
2) Line 55-58: Please avoid the repetition of the materials and methods.
Answer. Redaction was modified.
3) Line 187 and Line 211: Please provide appropriate references.
Answer. Done.
4) Line 225: The opening of the line should be changed.
Answer. Redaction was modified.
5) Line 229-230: Please cite an appropriate source and it might be helpful to compare this characteristic with other species of Stenamoeba to justify the claim.
Answer. According to the original version, as far as I am concerned, it refers to the last four lines of the results section, so I also do not understand why you are requesting an appropriate source.
6) Line 234: What is the source? Cite an appropriate source please.
Answer. Redaction was modified.
7) Line 306: It should be section 5.1
Answer. Corrected numbering
8) Line 321: Please change to 5.2
Answer. Corrected numbering
9) Line 333: Please provide a web link for the Primer3Pius software.
Answer. Link is in the place indicated
10) Line 336: Same as the previous comment (provide a web link).
Answer. Link is in the place indicated
11) Line 348: Please provide the name of the developer and a weblink for the MEGAX software.
Answer. Developer and link are in the place indicated.
12) Line 350: Please cite an appropriate source.
Answer. Reference added
We appreciate the reviewer's comments, which were very helpful in improving the quality of the manuscript.
Sincerely yours,
- FERNANDO LARES VILLA
Corresponding author
E-mail: flares@itson.edu.mx

Reviewer 2 Report
The main objective of this manuscript is to propose a new species of the genus Stenamoeba by describing two amoeba sp. nov. isolates. The article is clear and well written. Conclusions reported here are supported. Overall, this is a nice paper and will provide important contributions to the field.
I have minor improvements which should be made and that can be dealt with by additions to the text and without any new experiments.
Line 51: Too many commas. Commas after well-differentiated and genetically are unneeded.
Line 63: Unclear, please rephrase.
Lines 74-75: All data must be available. Where is the data for the average length and width of the motile shape? How many were measured? How was it done? There is nothing in the materials and methods.
Line 116-117: Unclear. Is M33 missing the first 53 bases due to sequencing problems or missing because they are deleted from M33 18S rRNA sequence?
Line 149: Missing word. This is “a” hypervariable …
Line 150: Change occur to occurs
Author Response
I am sending the revised manuscript No. Pathogens-844572, titled Stenamoeba dejonckheerei sp. nov., a Free-Living Amoeba Isolated from a Thermal Spring, once the reviewers suggestions were attended point by point as can be read in the following lines.
To Reviewer 2
1) Line 51: Too many commas. Commas after well-differentiated and genetically are unneeded.
Answer. Redaction was modified.
2) Line 63: Unclear, please rephrase.
Answer. Redaction was modified.
3) Lines 74-75: All data must be available. Where is the data for the average length and width of the motile shape? How many were measured? How was it done? There is nothing in the
materials and methods.
Answer. Redaction was modified including the light microscopic characterization in the material and methods section, measurements in results section and their discussion.
4) Line 116-117: Unclear. Is M33 missing the first 53 bases due to sequencing problems or missing because they are deleted from M33 18S rRNA sequence?
Answer. Redaction was modified adding the phrase “due to sequencing problems”.
5) Line 149: Missing word. This is "a" hypervariable ...
Answer. Redaction was modified.
6) Line 150: Change occur to occurs
Answer. Redaction was modified.
We appreciate the reviewer's comments, which were very helpful in improving the quality of the manuscript.
Sincerely yours,
- FERNANDO LARES VILLA
Corresponding author
E-mail: flares@itson.edu.mx

Reviewer 3 Report
The ms represents a description of a new thermotolerant amoeba species belonging to the genus Stenamoeba. This is the first finding of a thermotolerant member of this amoeba genus, and this finding expands the range of amoebae which potentially may be dangerous for human or become a vectors for various human infections. This is an evident importance of this finding. Molecular data leave no doubts for the status of this isolate as a new member of the genus Stenamoeba. The description is very detailed and carefully done.
I have several comments to the ms which (to my opinion) may help to improve the descriptions. First, general comments:
My main concern in the ms is the quality of the LM plate (Figure 1) which (I am sorry to say) is not representative and does not provide any visual information required to a specialist. No real locomotive form is shown in either of images. The overall quality of the plate is relatively low; even considering that authors a e limited with the simple upright microscope, it probably was not properly adjusted (aperture diaphragm and/or position of the condenser was wrongly set) and the focal plane is mostly not on the cell. In the image of the cyst almost no details are visible, even the cyst wall is not clear. These images should be replaced using properly Keller-adjusted microscope or (preferable) DIC or Phaco microscope. It is necessary to provide a figure of a flattened cell adhered to the glass and properly moving. It would be nice to provide not just a one photo, but a plate, showing a set of conformations of the locomotive form, so that the plate could provide a general impression about this species, showing what one might to expect when he is looking on this species.
I am not sure that nowadays it makes sense to provide image of an electrophoretic gel (Figure 2) - it is a technical part. Certainly if sequencing was successful, the gel has also been good.
Table 3 is hardly readable and provides details on the genetic distances, maybe it is rational to move it to additional materials. It makes sense to cut the right part (over the diagonal), because it mirrors the lower part (this is a usual practice for this kind of tables).
Overall, authors differentiate a new species basing almost exclusively on molecular data and made no attempt to consider the morphology, which is very briefly described. In the Discussion section, there is a reference to the forthcoming "more extensive micrographic study of the isolates will be left for later documentation once the images obtained by electron microscopy are available". This makes the description incomplete. Authors should probably either provide all data required to describe and differentiate a species or postpone the description until the entire dataset is available. Otherwise there are big chances that the diagnosis will require revision after more detailed study.
Specific comments:
Line 47: new genus, but not a new species (Platyamoeba stenopodia was the original name).
Lines 60-61: The genus Platyamoeba was abandoned by Smirnov et al. in 2007 (merged with Vannella), so it is better to say something like "former genus Platyamoeba (now as Vannella)". I should say that it is relatively hard to see real differences between the former Platyamoeba and Vannella in details of the locomotive form; the difference is usually clearer in the floating form. The suggestion made in these lines is probably based on the assumption of Stenamoeba as a former Platyamoeba.
Line 69 and legend to the Figure 1. May I suggest that "limax" shape is not an adequate description of Stenamoeba, which is rather flattened. I assume that the initial meaning of this term was exactly as authors mean (Ripella platypodia was described as limax amoeba), but this was the time when the 3D configuration of the cell was almost not taken into account. Nowadays the term "limax" is rather used to describe hartmannellids or heteroloboseans. Legend to the Fig. 1 - it is not clear, which isolate was illustrated.
Lines 74-75: The number of specimens measured (n=) should be indicated here or in material and methods section.
Overall, the LM description of a new species is very brief. It would be useful to know the details of the locomotive form like the relative size of the frontal hyaline area, presence of dorsal fold or ridges, the structure and the size of the nucleus, the details on the cytoplasmic inclusion. Not data on the floating form are provided. The description in the present form looks incomplete.
Lines 77-89: those are Methods rather than results. The only result here is the total length of sequence.
Lines 87-117: I am not sure it makes sense to provide all this technical details. The lack of sequencer capacity is like a broken lamp in the microscope - you just replace it, and there is not need to explain the whole story like "but I have not found a replacement it in my tablebox and had to order a new one, but it was promised to arrive in a week, so I had to go to the nearest WallMart and to buy a new one". May I suggest to skip all these technical details.
Line 114: here authors refer to a "Table 1 of Materials and Methods", however right there is another Table 1, belonging to Results. I assume that all table must have the throughout numeration (?).
Line 171: it is stated that "The tree has been rooted 171 using the sequence of Paradermamoeba levis (JN247435)" but figure 5 does not contain a root taxon.
Lines 172-173: the phrase "Stenamoeba is classified within the 172 Thecamoebida [3,4], but shows substantial differentiation from other thecamoebid taxa" is not clear here. What does it means in the description of the results of the analysis.
Line 163-170: Those are methods. It is possible to use the way that authors applied, and the results look reliable, but I cannot understand the reason for such overcomplication of the pipeline. Most of ML programs consider gaps as "N", why there is a need to perform analyses separately. Why RaxMl, which is almost a standard in this field was not applied?
Lines 193-195: values of the sequence identity can be compared only in complete sequences, not in the compressed alignment. Values made on compressed alignment depends on the alignment and cannot be absolute (because an alignment can vary).
Lines 214-223 - these are interesting data, but they do not belong to the present paper.
Lines 245-258 - those are mostly methods and is a partial repeat.
The diagnosis should not contain references to figures. It contains trivial description, which more corresponds to the genus. The only species-specific data here are the size data, but even the size of the nucleus is not provided. This diagnosis leaves almost no chance to differentiate a new species (except to the sequence data). May I suggest to add more details when the morphological description is improved.
The type material (if strain) should be deposited to one of internationally recognised culture collections (CCAP,ATTC, etc.) or provided as a satined preparation and (again) deposited to a recognised collection (NHM, etc.).
It is desirable to provide a comparison with congeners to show differences of the new species (or stress the absence of reliable differences).
Overall, this is a very interesting and promising study, but the ms requires certain upgrade and clarification. I am sure that it is possible to reach the proper quality of a species description and provide necessary data to make the description representative and highly useful for a community.
Author Response
I am sending the revised manuscript No. Pathogens-844572, titled Stenamoeba dejonckheerei sp. nov., a Free-Living Amoeba Isolated from a Thermal Spring, once the reviewers suggestions were attended point by point as can be read in the following lines.
To Reviewer 3
1) Line 47: new genus, but not a new species (Platyamoeba stenopodia was the original name).
Answer. Redaction was modified.
2) Lines 60-61: The genus Platyamoeba was abandoned by Smirnov et al. in 2007 (merged with Vannella), so it is better to say something like "former genus Platyamoeba (now as
Vannella)". I should say that it is relatively hard to see real differences between the former Platyamoeba and Vannella in details of the locomotive form; the difference is usually clearer in the floating form. The suggestion made in these lines is probably based on the assumption of Stenamoeba as a former Platyamoeba.
Answer. The ideas expressed in these lines were eliminated due to the re-edition of the results section, eliminating the repeated part of materials and methods (as a suggestion from a reviewer).
3). Line 69 and legend to the Figure 1. May I suggest that "limax" shape is not an adequate description of Stenamoeba, which is rather flattened. I assume that the initial meaning of this term was exactly as authors mean (Ripella platypodia was described as limax amoeba), but this was the time when the 3D configuration of the cell was almost not taken into account.
Nowadays the term "limax" is rather used to describe hartmannellids or heteroloboseans. Legend to the Fig. 1 - it is not clear which isolate was illustrated.
Answer. I respect your opinion on the use of the term "Limax", but I think that the term does not refer to the thickness of the amoeba, but to the appearance it has during movement. According to a dictionary definition, the term refers to the shape of the amoeba with a single wide, rounded anterior pseudopod and slow forward motion. https://www.merriam-webster.com/dictionary/limax
4) Lines 74-75: The number of specimens measured (n=) should be indicated here or in material and methods section.
Overall, the LM description of a new species is very brief. It would be useful to know the details of the locomotive form like the relative size of the frontal hyaline area, presence of dorsal fold or ridges, the structure and the size of the nucleus, the details on the cytoplasmic inclusion. Not data on the floating form are provided. The description in the present form looks
incomplete.
Answer. Redaction was modified including the light microscopic characterization in the material and methods section, measurements in results section and their discussion. Unfortunately, at this time we do not have a microscope with a Nomarski illumination system to detect the presence of dorsal folds or ridges, nor an electron microscope to know details about cytoplasmic inclusions, but it will be a priority to continue with the morphophysiological characterization of the new species.
5) Lines 77-89: those are Methods rather than results. The only result here is the total length of sequence.
Answer. Redaction was modified.
6) Lines 87-117: I am not sure it makes sense to provide all this technical details. The lack of sequencer capacity is like a broken lamp in the microscope - you just replace it, and there is not
need to explain the whole story like "but I have not found a replacement it in my tablebox and had to order a new one, but it was promised to arrive in a week, so I had to go to the nearest
WallMart and to buy a new one". May I suggest to skip all these technical details.
Answer. Once again we respect your opinion but precisely for this work, we believe that it is important to detail the steps that led us to determine without a doubt that it was a new species, especially when the GenBank is full of information about partial sequences or incomplete.
7) Line 114: here authors refer to a "Table 1 of Materials and Methods", however right there is another Table 1, belonging to Results. I assume that all table must have the throughout numeration (?).
Answer. Corrected numbering
8) Line 171: it is stated that "The tree has been rooted 171 using the sequence of Paradermamoeba levis (JN247435)" but figure 5 does not contain a root taxon.
Answer. A note was added in the figure title, on the external group of the phylogenetic tree indicating that it is not included due to the remoteness of the species under study.
9) Lines 172-173: the phrase "Stenamoeba is classified within the 172 Thecamoebida [3,4], but shows substantial differentiation from other thecamoebid taxa" is not clear here. What does it
means in the description of the results of the analysis.
Answer. Redaction was modified. The phrase was deleted because it was effectively out of the context of the results analysis.
10) Line 163-170: Those are methods. It is possible to use the way that authors applied, and the results look reliable, but I cannot understand the reason for such over complication of the pipeline. Most of ML programs consider gaps as "N", why there is a need to perform analyses separately. Why RaxMI, which is almost a standard in this field was not applied?
Answer. The reason is the experience and confidence in the program used. While RaxMl is used often, we are not aware that it is the standard. The appropriate ways of dealing with gap data also remains unsettled; it is unclear whether censoring all gap positions is more appropriate that assuming that a gap is a fifth state [N-A-T-C-G].
11) Lines 193-195: values of the sequence identity can be compared only in complete sequences, not in the compressed alignment. Values made on compressed alignment depends on the alignment and cannot be absolute (because an alignment can vary).
Answer. We prefer to maintain our description of sequence change. Values in the compressed alignment are less subject to variation. The values in the compressed versus non-compressed alignment were provided to indicate the degree to which gaps alter the perception of sequence divergence. Relative sequence divergence is sometimes used to indicate relative divergence time (i.e., the controversial concept of a molecular clock), but the occurrence of insertions or deletions do not follow the same underlying process as nucleotide substitution.
12) Lines 214-223- these are interesting data, but they do not belong to the present paper.
Answer. We disagree. These sequences indicate that other unrecognized taxa of Stenamoeba exist, and that they are also different from the new species. We feel that it is appropriate to mention them since they have not been recognized because they were observed in an environmental survey.
13) The diagnosis should not contain references to figures. It contains trivial description, which more corresponds to the genus. The only species-specific data here are the size data, but
even the size of the nucleus is not provided. This diagnosis leaves almost no chance to differentiate a new species (except to the sequence data). May I suggest to add more details when the morphological description is improved.
Answer. Redaction was modified including the light microscopic characterization in the material and methods section, measurements in results section and their discussion.
14) The type material (if strain) should be deposited to one of internationally recognized culture collections (CCAP, ATTC, etc.) or provided as a stained preparation and (again) deposited to a recognized collection (NHM, etc.).
Answer. Added phrase "there is currently no strain of available at either ATCC or CCAP" taken from an article describing new species.
15) It is desirable to provide a comparison with congeners to show differences of the new species (or stress the absence of reliable differences).
Answer. A comparison showing similarities and differences between the species can be found in the discussion section.
We appreciate the reviewer's comments, which were very helpful in improving the quality of the manuscript.
Sincerely yours,
- FERNANDO LARES VILLA
Corresponding author
E-mail: flares@itson.edu.mx

Round 2
Reviewer 3 Report
Thank you for detailed reply to my comments. May I note that while the most of replies and corrections are appropriate and have improved the ms a lot, in some points I still have to disagree.
Lines 35-38: these are methods.
Line 51: shortened name "S." should not be used as the first word in the sentence. Please use full name of the genus in this case.
Lines 61-63: looks like something dropped out here. What was the way to maintain amoebae. Have you worked with mixed cultures? This looks like a part of methods as well.
Line 63: in amoebae the hyaloplasm covers entire cell as an external layer; probably you mean the "frontal hyaline area".
Line 69: "The uroid" = "the posterior part of an amoeba". Uroidal structures means the presence of something differentiated on the posterior end of the cell. It looks (Fig. 1) as you have the uroid of bulbous type.
Line 78-79: an ostiole is labelled on the figure but not mentioned in the text. This is unusual for Stenamoeba, are you sure this is the structure?
Figure1. I am sorry to say, but the quality of LM plate remains far under any appropriate level. Geometrical deformation (horizontal) of all images is evident, an this makes false impression on the proportion of the cell. A plate like this in NOT acceptable for publication. I fully understand that DIC of Phaco may not be available, but please provide proper bright-field images. Images J-M - this is not a differentiated floating form, this is just a detached cell. Image S: I hardly can see, but if it is a structure, this is an uroid of bulbous type, with contractile vacuole inside (usual thing for Stenamoeba).
Figure 5. The outgroop must be on the image, or call the tree "unrooted". You may cut and shorten the branch leading to the root taxon, if it does not fit the figure (this is the usual practice).
Line 316: not "censored" but "compressed").
Line 374: correct "limacine". Please check across the entire text.
Line 391-392: repeat
Lines 435-437: the culture in you laboratory cannot be considered as a type material (see ICZN). I fully understand that it is a usual (bad) practice in amoebae research to have a type culture in some of the recognised collections (which are getting lost too often), but please, if (while) you have culture, make a stained preparation and deposit it to the British Museum of Natural History or some other collection like this, which is able to send them on request to be examined by other researchers. In other case you species has to be neotypificated in future.
Line 499: here you write that you have a microscope equipped with Phaco, but where are phase-contrast images??
Line 501: you use video stills to illustrate the LM morphology of amoebae. Please pay attention to the technically correct capturing of frames, including the proportions of the image. To the moment you have violated proportions in the Figure 1.
May I also note that I cannot agree with your consideration of the comparison of compressed alignments. There is is not repetitive universal algorithm of compression, you do this arbitrary. You are correct that values show less variations, but those are artificial digits, which do not reflect a real distance between sequences (which necessarily includes indels).
Author Response
Response to Reviewer
1) Lines 35-38: these are methods.
Answer. Redaction was modified
2) Line 51: shortened name "S." should not be used as the first word in the sentence. Please use full name of the genus in this case.
Answer. Redaction was modified
Lines 61-63: looks like something dropped out here. What was the way to maintain amoebae? Have you worked with mixed cultures? This looks like a part of methods as well.
Answer. Redaction was modified
Line 63: in amoebae the hyaloplasm covers entire cell as an external layer; probably you mean the "frontal hyaline area".
Answer. Redaction was modified
Line 69: "The uroid" = "the posterior part of an amoeba". Uroidal structures means the presence of something differentiated on the posterior end of the cell. It looks (Fig. 1) as you have the uroid of bulbous type.
Answer. We agree with you regarding the type of uroid, if bulbous also means morulate "according to Page's classification (1988), but honestly it was difficult to identify what type, and because of that in the discussion we refer to "an uroid-like structure in the previous region"
Line 78-79: an ostiole is labelled on the figure but not mentioned in the text. This is unusual for Stenamoeba, are you sure this is the structure?
Answer. Thanks to your observation, it reminded us that we should be more cautious in a report, until we have conclusive evidence, so we decided to remove the photo and only leave comments in the results and discussion sections.
Figure1. I am sorry to say, but the quality of LM plate remains far under any appropriate level. Geometrical deformation (horizontal) of all images is evident, and this makes false impression on the proportion of the cell. A plate like this in NOT acceptable for publication. I fully understand that DIC of Phaco may not be available, but please provide proper bright-field images. Images J-M - this is not a differentiated floating form; this is just a detached cell. Image S: I hardly can see, but if it is a structure, this is an uroid of bulbous type, with contractile vacuole inside (usual thing for Stenamoeba).
Answer. We recognize that trying to make some structures are better observed and adapted to the available space, we modified the shape of some, not all, but now all photos were all reduced respecting the proportions, as can be seen in the proportion of the sizes of the cysts and trophozoites. We repeated the test to have new photos of floating forms and we obtained two of the forms most similar to those described for S. sardiniensis (Geisen et al., 2014), which is different for S. polymorpha (Peglar et al., 2016), no floating form is described for S. stenopodia (Patsyuk, 2012), floating form with radiant pseudopodia blunt, in S. amazonica and S. limacina and the article does not show photos (Dykova et al., 2010) and "Floating form with several long, thin, mostly hyaline, irradiated but not pointed pseudopods" for Platyamoeba stenopodia (Page 1969), and last description is schematic. We believe that in our isolates the emerging bifurcating pseudopods is an interesting feature not yet described in Stenamoeba spp.
Figure 5. The outgroop must be on the image, or call the tree "unrooted". You may cut and shorten the branch leading to the root taxon, if it does not fit the figure (this is the usual practice).
Answer. We edited the tree and included the outgroup as recommended.
Line 316: not "censored" but "compressed").
Answer. To be consistent with last descriptions we eliminate the first line and we left information useful for futures molecular comparisons.
Line 374: correct "limacine". Please check across the entire text.
Answer. Done.
Line 391-392: repeat
Answer. We do not see any repeated line.
Lines 435-437: the culture in your laboratory cannot be considered as a type material (see ICZN). I fully understand that it is a usual (bad) practice in amoebae research to have a type culture in some of the recognized collections (which are getting lost too often), but please, if (while) you have culture, make a stained preparation and deposit it to the British Museum of Natural History or some other collection like this, which is able to send them on request to be examined by other researchers. In other case your species has to be neotypificated in future.
Answer. We are more interested in placing the strains in an international collection, but at the moment and due to the COVID-19 pandemic it is very complicated, but I assure you that we will send them as soon as possible and for the moment strains are available at the present address.
Line 499: here you write that you have a microscope equipped with Phaco, but where are phase-contrast images??
Answer. Due to your critical welcomes and that we are not sure of the efficiency of the phase contrast operation, we decided to leave the indication that the photos were taken with a light microscope.
Line 501: you use video stills to illustrate the LM morphology of amoebae. Please pay attention to the technically correct capturing of frames, including the proportions of the image. To the moment you have violated proportions in the Figure 1.
Answer. all photos were all reduced respecting the proportions.
May I also note that I cannot agree with your consideration of the comparison of compressed alignments. There is not repetitive universal algorithm of compression, you do this arbitrary. You are correct that values show less variations, but those are artificial digits, which do not reflect a real distance between sequences (which necessarily includes indels).
Answer. We also respect your point of view.
We appreciate your comments that were very useful to improve the quality of the manuscript
